**Temporary stratification promotes large greenhouse gas emissions in a shallow eutrophic lake**
Thomas A Davidson[1, 2], Martin Søndergaard[1, 2, 3], Joachim Audet[1, 2], Eti Levi[1], Chiara Esposito[1, 2], Tuba
Bucak Onay[1], Anders Nielsen[1,4].
[1] Lake Ecology, Department of Ecoscience, Aarhus University, Denmark
[2] WATEC Aarhus University Centre for Water Technology, Aarhus University, Denmark
[3.] Sino-Danish Centre for Education and Research (SDC), Beijing, China
[4.] WaterITech Aps, Døjsøvej 1, 8660 Skanderborg, Denmark
Corresponding author: Thomas A Davidson, Department of Ecoscience, Aarhus University, C. F. Møllers
Alle 4-6, DK-8000 Aarhus C, Denmark, e-mail: thd@ecos.au.dk

## Abstract

Shallow lakes and ponds undergo frequent temporary thermal stratification. How this affects greenhouse gas (GHG) emissions is moot, with both increased and reduced GHG emissions hypothesised. Here, weekly estimation of GHG emissions, over growing season from May to September, were combined with temperature and oxygen profiles of an 11 hectare temperate shallow lake to investigate how thermal stratification shapes GHG emissions. There were three main stratification periods with profound anoxia occurring in the bottom waters upon isolation from the atmosphere. Average diffusive emissions of methane ($CH_4$) and nitrous oxide ($N_2O$) were larger and more variable in the stratified phase, whereas carbon dioxide ($CO_2$) was on average lower, though these differences were not statistically significant. In contrast, there was a significant, order of magnitude, increase in $CH_4$ ebullition in the stratified phase. Furthermore, at the end of the period of stratification, there was a large efflux of $CH_4$ and $CO_2$ as the lake mixed. Two relatively isolated turnover events were estimated to have released the majority of the $CH_4$ emitted between May and September. These results demonstrate how stratification patterns can shape GHG emissions and highlight the role of turnover emissions and the need for high frequency measurements of GHG emission which are required to accurately characterise emissions, particularly from temporarily stratifying lakes.

Keywords: Climate change; lake stratification; methane; carbon dioxide; nitrous oxide; climate feedbacks

## 1. Introduction

Fresh waters are key sites for the processing of greenhouse gases (GHG), methane ($CH_4$), carbon dioxide ($CO_2$) and nitrous oxide ($N_2O$). Shallow lakes, in particular, have been identified as hot spots of $CH_4$ release, particularly when ebullition is taken into account (Davidson et al., 2018; Aben et al., 2017). The certainty that fresh waters are large emitters of GHGs contrasts with the uncertainties associated with the quantities emitted and this is in large part due to historical paucity of measurements (Cole, 2013). A recent study identified the highly variable emissions from lakes and ponds (Rosentreter et al., 2021). Whilst different morphometric features and chlorophyll-a explained some of the emission patterns (Deemer and Holgerson, 2021), it is also clear that a dearth of measurement combined with these highly variable emissions makes determining the drivers and controls of those emissions a challenge, which in turn makes predicting future emissions difficult.

The current and future effects of climate change on lakes in general and on their GHG emissions are relevant questions as there is potential for positive feedbacks and synergies with other human impacts such as eutrophication (Davidson et al., 2018; Beaulieu et al., 2019; Delsontro et al., 2016; Meerhoff et al., 2022). Taking a broad metabolic theory of ecology approach, temperature increases should promote methanogenesis and shift the balance from primary production to respiration increasing $CO_2$ emission at cellular and ecosystem scale (Yvon-Durocher et al., 2010). However, empirical and experimental data indicate that temperature is not the sole control of primary production and methanogenesis. In particular, eutrophication, and the promotion of large algal crop, has been associated with increased emissions of $CH_4$ and $N_2O$ (Delsontro et al., 2016) both by diffusion and ebullition (Zhou et al., 2019). Furthermore, in what is globally the most abundant lake type, small shallow lakes, where macrophytes can colonise large areas of the lake bed, trophic state and the dominance of submerged plants or algae may be more important than temperature in shaping GHG dynamics (Davidson et al., 2015; Davidson et al., 2018; Bastviken et al., 2023).

62

Climate change effects on lakes are not limited to increases in average temperatures and lengthening of

the growing season. Increases in both the frequency and intensity of heat waves are predicted, which will

promote the warming of surface waters and in turn make permanent and temporary thermal stratification

of lakes more likely (Woolway and Merchant, 2019), even in lakes typically classified as non-stratifying

(Kirillin and Shatwell, 2016). A recent study Holgerson et al. (2022) identified stratification and mixing

patterns in small water bodies, with permanent summer stratification common and frequent mixing

occurring in larger standing waters (>4 ha) lakes. Such periods of stratification and mixing events are

likely to have profound effects on GHG dynamics. Emissions of gases, in particular $CH_4$, that accumulate

in the isolated bottom waters of a stratified lake, occurs upon mixing and can make very significant

contributions to cumulative emissions (Schubert et al., 2012). High-resolution studies of sites that

undergo temporary stratification are rare. Though Søndergaard et al. (2023b), recently showed how

stratification shapes patterns and processes across the entire ecosystem, including short term effects on

dissolved GHG concentration in bottom and surface waters. In terms of its effects on GHG dynamics,

there are potentially antagonistic processes at work in a stratified lake. On the one hand the 'shield effect'

results in lower temperatures at the sediment surface slowing down metabolic processes that scale with

temperature, i.e. methanogenesis and mineralization of organic carbon (C), reducing emission and

promoting C burial. On the other hand, anoxia at the sediment surface may shift processes towards

fermentation, increasing the proportion and total amount of $CH_4$ produced and perhaps reducing C burial

(Bartosiewicz et al., 2019). Recent work combining empirical observations and models has suggested that

shielding effects are larger than the anoxia effects and that stratification, in general, increases C burial and

reduces GHG emissions  (Bartosiewicz et al., 2015). The stratification induced isolation of bottom waters

was reported to lead to reduced ebullition of $CH_4$ and a shift to diffusive pathways (Bartosiewicz et al.,

2015. It might, however, be predicted that in shallow lakes stratification would lead to much larger $CH_4$

release as anoxic conditions would limit $CH_4$ oxidation by $CH_4$ oxidizing bacteria (MOBs) (Bastviken et

al., 2008). There may also be other factors with the potential to increase GHG emission, such as sediment

organic content and lake trophic status (Delsontro et al., 2016), which may interact with stratification
patterns in shaping GHG emissions.

In this study, we used data from a shallow lake with high frequency measurements of temperature profiles
combined with weekly measurements of dissolved gas concentrations in the surface and bottom waters
and continuous measurement of ebullitive emissions of $CH_4$ to track the effects of lake stratification on
GHG emissions. The key question was how ebullitive and diffusive fluxes of the key GHGs: $CH_4$, $CO_2$
and $N_2O$ respond to temporary thermal stratification.

## 97    2.  Materials and methods.

### 98    2.1 Study site

Ormstrup lake, located in Denmark (lat 56.326°, lon 9.639°) (Fig.1) (depth map with GHG sampling
locations), is an 11 ha, shallow lake (average depth 3.4 m), with a maximum depth of 5.5 m, and with a
relatively long hydraulic retention time (> 1 year). The lake is eutrophic with high TP and chlorophyll-a
(Table 1; Søndergaard et al., 2022) with very sparse occurrence of submerged plants.

### 104   2.2 Depth profiling and high frequency measurements

In June 2020, a Nexsens (NexSens Technology, Fairborn, OH, USA) CB-450 data buoy system
(https://www.nexsens.com/pdf/CB450_datasheet.pdf) was deployed at the deepest point of the lake
equipped with a Nexsens TS210 thermistor string https://www.nexsens.com/pdf/TS210_datasheet.pdf)
with temperature nodes measuring at 4 levels; one sensor "in air", ca. 5 cm above the water surface, (but
shielded from direct light), and three sensors at -1, -2, -3 meters, respectively relative to the water surface.
In addition two Aqua TROLL 500 (In-Situ, Fort Collins, CO, USA) multi-sondes were mounted near the
surface (-1.0 meters) and at deeper water depth (-3.8 meters). The near surface and deeper water sonde
were configured with sensors to measure dissolved oxygen (DO) and water temperature (Tw). The optical
sensors were calibrated according to manufacture guidelines and checked on a weekly basis.

The optical sensors of the Aqua TROLL 500 have a built-in wiper mechanism to clean sensor heads to
hamper bio-fouling. The wiper function was enabled to perform cleaning in sync with sensor
measurements, hence every 15 minutes. In addition, manual cleaning of sensor heads was done every
week, while routine manual field monitoring was carried out at the lake.  Prior to the deployment of the
buoy, and as a validation exercise for the buoy data, weekly manual profiles of DO and Tw were collected
at the deepest point.

Periods of stratification and depth of the thermocline were defined using the r package rlakeanalyzer
(Winslow et al., 2019) based on the density gradient of the water column from the weekly manual
profiling of the system. During periods of defined as stratified, there were partial mixing events where the
depth of the thermocline changed and there was some mixing of the sub epilimnetic water and the surface
waters, whilst the bottom waters below 3.5 metres remained undisturbed.

**2.3 Water chemistry**
Water samples for the analysis of Chlorophyll-a were collected weekly from the 20. April 2020 from
surface (-0.5 m) water at station 3 (Fig. (Søndergaard et al., 2005). A volume of water ranging from (0.2
to 1 litre) was filtered and the GFC papers preserved for chlorophyll-a analysis, which were determined
spectrophotometrically after ethanol extraction (Jespersen and Christoffersen, 1987) and alkalinity was
measured weekly by gran titration (Søndergaard et al., 2005).  Depth profiles of temperature, electrical
conductivity (EC) and dissolved oxygen (DO) were measured manually with an Aqua TROLL 500 probe
from every -0.5 or -1 m down to -5 m depth).

**2.4 Greenhouse gas sampling**

2.4.1 Dissolved concentration

Samples of dissolved concentrations of $CH_4$, $CO_2$ and $N_2O$ were collected weekly from the 20. April 2020 from surface waters and weekly from surface and bottom water from the 26. May 2020 to the 13. October 2020. The samples were taken using head-space equilibration after (Mcauliffe, 1971), where 20 ml of water was collected from just below the water surface and 20 ml of $N_2$ was introduced as a headspace in a 60-ml syringe and then shaken vigorously for one minute. The 20 ml headspace was then transferred to a 12-ml pre evacuated glass vial.

Gas concentrations in the headspace were determined on a dual-inlet Agilent 7890 GC system interfaced with a CTC CombiPal autosampler (Agilent, Nærum, Denmark) (Petersen et al., 2012). For the GC, certified $CO_2$, $CH_4$ and $N_2O$ standards were used for calibration and validation. Aqueous concentrations in $N_2O$, $CH_4$ and $CO_2$ were calculated from the headspace gas concentrations according to Henry's law and using Henry's constant corrected for temperature and salinity (Weiss, 1974; Weiss and Price, 1980; Wiesenburg and Guinasso, 1979). A recent study (Koschorreck et al., 2021) identified significant bias in the estimate of $CO_2$ concentrations using headspace equilibration at lower concentrations. We applied their correction using separately measured alkalinity as described in Koschorreck et al. (2021).

The fluxes of $N_2O$, $CH_4$ and $CO_2$ between the water and the overlying atmosphere were estimated as

$$f_g = k_g\left(C_{wat,g} - C_{eq,g}\right)$$

Where $f_g$ is the flux of a specific gas $g$, $k_g$ is the piston velocity of the gas and $C_{wat,g} - C_{eq,g}$ is the gradient of concentration between the concentration of gas dissolved in the water ($C_{wat,g}$) and the concentration of gas the water would have at equilibrium with the atmosphere ($C_{eq,g}$).

We calculated a gas transfer velocity $k_{600}$ for each sampling occasion using the relationship based on windspeed described in (Cole and Caraco, 1998).

$$k_{600} = 2.07 + 0.215U_{10}^{1.7}$$
$U_{10}$ is the mean daily windspeed at 10m (m s$^{-1}$) obtained from the Danish meteorological institute
(DMI;20x20 km grid data)


$$k_g = k_{600} \left(\frac{Sc_g}{600}\right)^x$$

$Sc_g$ is the Schmidt number(Wanninkhof, 1992) of the specific gas $g$. We chose $x$ =-2/3 as this factor is
used for smooth liquid surface (Deacon, 1981).

Daily flux rates were calculated using linear interpolation of the weekly surface measurements from each
of the sampling points. The diffusive surface water fluxes were calculated by taking an average of the daily
flux rate from the 12. May 2020 to the 13. October 2020 for each location. Then an average of the 3 locations
was multiplied by the area of the lake and the number of days covered by the study, here 126 days was
chosen to match the period over which ebullition was measured.

The total content of the gases in the lake's bottom waters were calculated from the dissolved
concentration of the gases multiplied by an estimate of the volume of the water in the hypolimnion. The
volume of water in the hypolimnion was estimated from the lake profiles manually conducted on the day
of sampling. The top of the hypolimnion was determined by the depth below which oxygen was less than
0.5 mg l$^{-1}$. A detailed bathymetry of the lake allows the calculation of the area and therefore volume of
water that lies below a given depth.

During the study period two major turnover events occurred, the process of lake turnover and full mixing
can take a number of days, and the outgassing even longer. The oxygen data, from the buoy, indicated
that it can take up to four days and this provides time for $CH_4$ oxidation to occur (Søndergaard et al.,
2023b). In order to estimate the amount of $CH_4$ oxidised over the course of the multiple days of degassing
we directly measured $CH_4$ oxidation rates in the surface waters of the lake. This was done in June 2023 in
five locations in this lake using methods outlined in  (Thottathil et al., 2019) where five water samples
from five different locations and each was incubated over 4 days with and the change in $CH_4$
concentration used to calculate oxidation rates. We used the minimum (0.267 µg $CH_4$-C $l^{-1}$ $h^{-1}$) , mean
(0.44 µg $CH_4$-C $l^{-1}$ $h^{-1}$) and maximum (0.58 µg $CH_4$-C $l^{-1}$ $h^{-1}$) oxidation rates to estimate the range of $CH_4$
oxidation likely to have occurred over the course of the two main turnover events. Assuming that the
degassing took four days, these rates would consume between 2 and 8% of the $CH_4$ contained in the
hypolimnion. Using the mean oxidation value the turnover fluxes were reduced by 4.1% on the 30$^{th}$ of
June 2020 and by 6% for the 25$^{th}$ August 2022.

2.4.2 Ebullition
The ebullitive flux of $CH_4$ was estimated using at total of 40 floating chambers placed on 4 transects of 10
chambers each (Fig. 1). The chambers have a volume of 8 litre and a surface area of 0.075 m$^2$, similar to
those used by (Bastviken et al., 2015). As the existing literature indicated that ebullition is lower as water
depth increases (Wik et al., 2013) the transects were placed to maximise the measurement of the low end
of the depth gradient on the shallower slopes of the western end of the lake (Fig. 1). The average and
maximum depth of each transect was T1: 293 cm and 472 cm; T2: 181 cm and 267, T3: 223 cm and 300
cm and T4 166 cm and 220 cm.The chambers were set on the 14. May 2020 and sampled every two
weeks from that date, and on one occasion after one week until September 17$^{th}$, which is a period of 127
days. Twenty ml of sample was taken from the floating chamber and injected into a pre-evacuated 12 ml
vial (exetainer, Labco). Gas concentrations were determined on the same GC than described above
(Petersen et al., 2012)
Ebullitive flux of $CH_4$ was estimated as:
$$\frac{p_{gas} \times Vol_{bub}}{t \times A}$$

Where $p_{gas}$ is the concentration of CH$_4$ in the gas that was trapped, $Vol_{bub}$ is the volume of the chamber
(i.e. 7L), $t$ is the time during which the samples was collected and $A$ is the area of chamber (i.e. 0.075 m$^2$).
A portion of the CH$_4$ released via ebullition in the chamber will have re-dissolved in the water or might
leak through the chamber walls, thus underestimating the ebullitive flux. We have made a number of
measurements to constrain this error and to compare estimates based on static chambers with other
approaches. The result show that whilst static chambers underestimate ebullition, given the temporal
variability of ebullition, static chambers continually deployed provide a better estimate of average ebullition
than short term (24-48 hours) deployment using portable gas monitors or flushing chambers.

Therefore, whilst static chambers method cannot be said to accurately quantify CH$_4$ emissions, they can be
relied upon to compare differences in ebullition between time periods, with the caveat that they are always
an underestimate of actual ebullitive flux.

Total ebullitive flux from the lake was calculated by taking a mean of the emissions from each transect over
the 126 day period. Then taking an average of the means of four transects and multiplying this by the time
of deployment of the chambers in days, which was 126 days, and by the area of the lake. This gives a total
ebullitive flux of CH$_4$ for the lake over the period of measurement from May to mid-September.

The three different flux types, surface diffusion, ebullition and turnover emission were then converted in
comparable units of total lakes emissions (as g or kg of gas) over the studied period and also converted into
CO$_2$-equivalents using a conversion factor related to their 100 year global warming potential (GWP) of 28
for CH$_4$ and 265 for N$_2$O.

**2.5 Statistical methods**
To test for a significant difference among the emissions from the stratified and mixed phase we used
generalised least squares (GLS) with a variance function to account for heterogeneity of variance between
the phases. In the case of the ebullitive flux, as the collected phase often covered periods including both
mixed and stratified phases there were three categories, mixed, stratified and both mixed and stratified.
All analysis was carried out in R version 4.2.1 (R Development Core Team, 2022) and the GLS used the
package nlme  (Pinheiro et al., 2014).

## 243 3.0 Results

### 244 3.1 Lake physical and chemical characteristics

Depth profiles measured weekly from April show that stratification was initiated by the 26. May 2020 this
may have broken down briefly and established again, visible in the temperature sensors for the buoy on
the 5. June 2020 (Fig. 2). There were then 12 days of mixing followed by stable period of stratification
with onset the 14. June 2020 and a duration of 16 days until a mixing event around the 30. June 2020. The
following two weeks had cooler water and a mixed water column, herafter a ca. 6 day period of
stratification from the 15. to 21. July 2020. A mixed phase of two weeks then followed until stratification
reestablished on 4. August 2020 and persisted until the end of August, partial mixing is indicated by the
buoy data from the 21. August 2020, but the weekly manual profile to deeper water inidcate that full
mixing did not occur until after the 25. August 2020. The effects of the stratification and mixing events on
the high frequency DO data measured at -3.8 m are clear, with rapid deoxygenation occuring after the
onset of stratification and oxic bottom waters returning when the lake mixed (Fig, 2).  The pattern in
chlorophyll-a also follow, to some degree, those of stratification, with the exception of early spring.
Chlorophyll-a values were extremely high in spring peaking at the start of June 2020 and falling gradually
(Fig. 2). (Søndergaard et al., 2023b)During the periods of stratification chlorophyll biomass was lower,
and when a mixing event occurred the values increased, which is particularly evident in the July mixing
periods (Fig. 2).

**3.2 Concentrations of dissolved gases and fluxes from the surface waters.**
The concentrations of the dissolved gases showed great variation from near or below atmospheric
concentrations in some cases and up to an extremely high concentration (over 5 mg $CH_4$ C $l^{-1}$) in the
bottom waters on the 30. June 2020. There was some spatial heterogeneity in the surface waters, with the
more littoral locations showing the greatest variation and the highest values (Figs. 3,4,5). In particular the
most littoral zone, where the water was shallower around 1 m in depth, showed the highest values just
prior to, or coincident with, the stratification turnover. Table 2 shows the mean diffusive flux of each gas
over the sampling period along with the mean flux in mixed and stratified phases. For $CO_2$ there was a lot
of temporal variation in flux dynamics, though not a large difference between mixed and stratified phases
in terms of mean values (Table 2). There were some periods of $CO_2$ influx in spring and later summer and
these tended to coincide with the end of a mixed phase and the start of the stratification phase. Nitrous
oxide concentrations were generally low (Figs 4 & 5) with the lake being a source of $N_2O$ in the spring
period and a sink or a very small source thereafter. The $CH_4$ concentration in the surface waters (Fig. 3)
and the calculated diffusive emissions are relatively low, but did increase in the stratification periods with
higher average values (Table 2 & Fig. 6). There was also some spatial variation with higher $CO_2$ and $CH_4$
diffusive emissions in the shallower sampling locations, both in stratified and mixed conditions (Fig. 6).

The most marked patterns in GHG concentration were evident in the bottom waters sampled at -4.5 m,
which accumulated to very large concentrations of $CO_2$ but particularly $CH_4$ in the periods of
stratification (Fig. 3 & 4). The ratio of $CO_2$ to $CH_4$ is illustrative in highlighting how stratification has
altered the biogeochemical processes in the hypolimnion with $CH_4$ production becoming more prevalent. .
For example on 30. June 2020 after 16 days of stratification the the ratio $CO_2$:$CH_4$ in the bottom waters
was 0.8, whereas 7 days later after the mixing event it was 187 at the same depth.

**3.3 Ebullitive fluxes**
The $CH_4$ bubble flux, presented here as mean values for each of the 4 transects, ranged from 0.303 to 81.1
mg $CH_4$ C $m^2$ $d^{-1}$ for the individual transect over the growing season measurement. There is a very clear,
statistically significant impact of stratification on the ebulltive efflux of $CH_4$ with stratified periods
showing significantly markedly higher levels of emission (Fig. 7 and Table 2). In addition, there was a
difference in average emissions among the different transects, with those with lower average water depth
(T2 & T4) having lower emission than the transects with chambers over deeper water (T1 & T3) (Fig. 7).
The samples collected from the chambers reflect two weeks of bubble and diffusion collection and the
quantification of the flux is therefore an average of the period of chamber deployment, which was two
weeks, or in one case a single week (Fig. 7). This two week period on occasion covered both stratified
and mixed phases and on these occassions efflux was intermediate between purely mixed and stratified
periods (Table 2 and Fig. 7).

**3.4 Total lake fluxes**
Scaling up the results to total flux of gases from the whole lake over the period of study and including the
estimated emissions from two turnover events show a very different effect of stratification on the balance
of types of emissions for the three gases. The majority of $CH_4$ emission (56%) result from the two short-
lived turnover events (Fig. 8), whereas their contribution to $CO_2$ and $N_2O$ emission was 5% and 1%
respectively.

Fluxes of $CO_2$ and $N_2O$ were mostly diffusive, which represented  95% of emissions of both gases.
Methane diffusive flux was 14% of total emission, whereas $CH_4$ ebulltion was more than twice as much at
29% of total $CH_4$ emission. In terms of global warming potential $CO_2$ and $CH_4$ emission were
comparable, but the contribution of the turnover efflux was the dominant factor for $CH_4$ emissions.

## 311    4. Discussion

This study set out to assess the role of thermal stratification on the GHG dynamics in a lake undergoing
frequent but temporary stratification.  We found that the emission of the three GHGs showed different
degrees of variation between the mixed and stratified phases. The largest and most significant variation
was in $CH_4$ ebullition (Table 2), whilst the difference in diffusive fluxes, though marked for $CH_4$ was not
significant. The mean of the total emissions from Ormstrup in the stratified phase (59.9 mg $CH_4$-C $m^{-2}$
$day^{-1}$) corresponds relatively closely to the mean of the total emissions (ebullition plus diffusion) reported
for lakes in this size range (47 mg $CH_4$-C $m^{-2}$ $day^{-1}$) from a paper synthesising multiple studies
(Rosentreter et al., 2021). The mean emissions for the whole period (26.6 $CH_4$ -C $m^{-2}$ $day^{-1}$ ) were lower
than Rosentreter et al. (2021) but similar to other studies with mean emissions of 30.9, 20.7 and 22.7 $CH_4$
-C $m^{-2}$ $day^{-1}$ and were reported by Peacock et al. (2021), Sø et al. (2023) and Peacock et al. (2019)
respectively. Whereas the average $CO_2$ (504 mg $CO_2$-C $m^{-2}$ day ) at Ormstrup was lower than  993.5 1 mg
$CO_2$-C $m^{-2}$ $day^{-1}$ measured by Peacock et al. (2021) but higher than the 264.6 and 205.1 mg $CO_2$-C $m^{-2}$
$day^{-1}$ measured by  Sø et al. (2023) and Peacock et al. (2019) respectively. The different temporal
resolution and duration of these studies, eleven single day sampling from April to December (Peacock et
al., 2021), five days continuous sampling on one occasion in late September (Sø et al., 2023) and a single
early summer snapshot (Peacock et al., 2019) make direct comparison difficult. The data here do,
however, provide a clear answer to the question of how thermal stratification effects GHG dynamics in
shallow eutrophic lakes with an increase in total emissions (diffusion, ebullition and turnover) during the
stratified period (Table 2, Fig 9). Previous work, combining observations and modelling suggested the
opposite patterns (Bartosiewicz et al., 2019) as the shielding effect of the stratification results in cooler
bottom waters which reduces $CH_4$ production due to the process being temperature dependent
(Bartosiewicz et al., 2016). This strong shielding effect may apply in deeper lakes experiencing more
stable stratification, or less eutrophic lakes. The result here from a relatively shallow eutrophic lake,
indicate that temporary stratification causes increases in GHG emissions. **4.1 Diffusive fluxes**Diffusive
emissions did not, on average, show a strong stratification effect (Table 2).  In particular variation in $N_2O$
emissions did not match patterns of stratification, with emissions more directly related to nitrate
concentrations (Audet et al., 2020), as reflected by the fact the lake is a sink of $N_2O$ in late summer when
nitrate was below detection limits for several weeks. There were peaks in emission of $CH_4$ and $CO_2$ at the
end of stratification periods, particularly in the shallower water sampling points (Fig. 6). There were
periods of influx of $CO_2$, which coincided somewhat with periods of stratification, but the pattern was not
consistent as other factors, for example, chlorophyll-a concentration also play a role.

Littoral zones can have markedly different GHG dynamics to deeper zones due to shallower water having
lower pressure (Wik et al., 2013), less time for $CH_4$ oxidation (Bastviken et al., 2008) or abundant plants
which influence a range of biogeochemical processes (Davidson et al., 2018; Esposito et al., 2023). It is
therefore possible that littoral zone dynamics could cause these differences. However, the increase
occurred at all three sampling points at the end of June 2020, which indicates a lake-wide driver and the
peak may represent the start of mixing after stratification. Strong winds were measured on the 29[th] and
30[th] June 2020 (Søndergaard et al., 2023b) coincident with these increased littoral emissions. These
winds would have caused lateral movement of the surface water causing an upwelling of bottom water,
rich in $CH_4$ and $CO_2$, in the littoral margins at the opposite end of the lake. Thus, whilst we do not have
direct evidence it seems more likely that these increased emissions in the littoral zone were driven, at least
in part, by the upwelling of GHG rich bottom waters.


**4.2 Ebullitive fluxes**
In contrast to the diffusive flux, the ebullitive emission of $CH_4$ shows a very clear response to
stratification with an order of magnitude difference in emissions between periods where the sampling
reflected purely mixed or stratified periods (Table 2 & Fig. 7). The two-week resolution of the sampling
meant that some samples covered both stratified and mixed phases and these samples had intermediate
fluxes, as they cover both low (mixed) and high emission (stratified) periods. The spatial variation in
ebullition is also illustrative of the impacts of stratification and the role of anoxia in shaping $CH_4$ fluxes.
The two transects with the largest mean and maximum depths (T1 and T3) had the largest emissions, with
the deeper of the two (T1) having the highest emissions and showing the largest relative increase during
the stratification phases. This pattern is different to that found some other studies where bubble emissions
were larger in shallower water (Wik et al., 2013), although in this, and another study (Sø et al., 2023),
there was in increase in bubble flux in deeper water in late summer. The deeper water at Ormstrup
experienced anoxia early in season resulting in locations with deeper water having higher ebullition rates
than shallower areas. This is at odds with ideas stemming from the metabolic theory of ecology stating
that temperature (Yvon-Durocher et al., 2014) in particular at the sediment surface (Bartosiewicz et al.,
2019) can be used to predict $CH_4$ efflux. Whilst $CH_4$ production is temperature dependent at the cellular
level, $CH_4$ emissions were rather independent of the sediment temperature, for example in the first two
weeks of July 2020 emissions were low and the sediment surface temperature was relatively high. Thus,
temperature alone is a poor predictor of ecosystem scale $CH_4$ emissions.

It should be noted that the methods used to estimate bubble flux here, where floating chambers are
sampled every two weeks is a "less than perfect method", which in nearly all cases will underestimate
ebullitive flux. Logistical and financial constraints make continual sampling difficult and here we
balanced these constrains against the greater time required to apply more accurate methods, such as
bubble traps (Wik et al., 2013), automatic flushing chambers (Bastviken et al., 2015). Such is the
variability of bubble flux in space and time that using measurement from a shorter period of 1-2 days can
result in a larger error in estimation of emissions than results from the longer term deployment of a static
chamber (see supplementary materials 1 and 2). The results in figure 7 show that sampling a single week
a year or even more regular monthly sampling of a shorter duration would be unlikely to accurately
characterise ebullition. Bubble traps have been used on longer terms but in eutrophic systems they can
suffer extensive biofouling which can impede their use. Thus, the continuous monitoring of ebullition
using static chamber with known biases was deemed the least worst method available, but we
acknowledge that ebullitive emissions are underestimated. We further acknowledge that this approach of
static chambers should, where possible, be replaced by other methods to estimate ebullition, such as
automatic flushing chambers. It is difficult to compare the mean values of emission with other studies as
there are different scales of measurement both in space and time. However, comparing the values for
ebullition recorded here with other longer-term studies carried out in lakes using bubble traps (Burke et
al., 2019; Delsontro et al., 2016), shows higher values recorded at Ormstrup lake compared with other
lakes,  but lower values that have been measured in ponds (Ray and Holgerson, 2023; Delsontro et al.,
2016), the latter being known to have higher emissions of $CH_4$ (Holgerson and Raymond, 2016).

**4.3 Turnover fluxes**
In addition to the diffusive and ebullitive emissions, the turnover flux, which consists of the gases
accumulated in the hypolimnion being released on turnover, was also estimated, with a correction of $CH_4$
oxidation applied. There were two major turnover events at the end of June and in late in August 2020,
which were preceded by 16 and 22 days of stratification, respectively. It was not possible to directly-
measure turnover flux, as they are relatively discrete events where the efflux likely occurs over the course
of a few hours, or a few days (Søndergaard et al., 2023b). Thus, the efflux estimation is based on a series
of assumptions and thus must be treated with caution. Notwithstanding this uncertainty, we can be
confident the turnover flux represents a very large proportion of the total emission of $CH_4$ emissions from
Ormstrup Lake over the growing season. We estimate it contributed more than 50 % of growing season
$CH_4$ emissions and 5 % of $CO_2$ emissions. This highlights a very significant, and difficult to measure,
contribution to GHG emissions from lakes undergoing temporary stratification, which are among the
most common lake type in Denmark (Søndergaard et al., 2023a).

**4.1 Stratification effects**
The results here suggest that GHG dynamics were driven both directly and indirectly by the stratification
patterns and the anoxia it induced in the bottom waters. At Ormstrup Lake the thermal stratification of the
water column quickly led to anoxia, with only a matter of hours to days for the oxygen to be consumed
once the bottom waters were isolated (Fig. 2). The ratios of $CO_2$:$CH_4$ evidence how this promotes $CH_4$
over $CO_2$ production in the stratification phase (see Fig 9). In addition to promoting $CH_4$ production such
conditions would preclude, or severely limit, oxic $CH_4$ oxidation, which has the potential to consume a
large proportion of $CH_4$ produced in the anoxic sediments (Bastviken et al., 2008), though anoxic
consumption of $CH_4$ can still occur (Blees et al., 2014).  The raw emission data do not provide any direct
information on the balance of production versus oxidation, but the $CO_2$:$CH_4$ suggest there was marked
shift to conditions where methanogenesis was the dominant process and there was reduced $CO_2$
production. Studies have shown that $CH_4$ oxidation can consume large proportions of the $CH_4$ produced
under hypoxia (Saarela et al., 2019) and it is possible that there is intense $CH_4$ oxidation occurring at the
thermocline during the periods of stratification at Ormstrup lake , but this was not directly measured at the
lake. In addition to the more direct effects of anoxia there may be some indirect effects of the patterns of
stratification and mixing that promote greater GHG emissions. Søndergaard et al. (2023b) recently
reported how nutrient dynamics at Ormstrup Lake were altered by the lake stratification and full details
can be found there, of relevance here is the impact on chlorophyll-a which saw a large spring peak after
which the abundance tracked the stratification and mixing regime, with a lag time. There was a general
reduction, or at least no increase as the stratification period progressed, perhaps due to nutrient limitation
in the epilimnion. Upon mixing there was generally an increase in chlorophyll-a, though the weekly
sampling resolution makes this difficult to assess. Chlorophyll-a and the labile dissolved organic carbon
(DOC) that result from abundant chlorophyll-a have been shown to be associated with higher diffusive
and ebullitive $CH_4$ emissions (Davidson et al., 2015; Beaulieu et al., 2019; West et al., 2012; Zhou et al.,
2019). It is not possible to say here whether a stable summer long stratification would have led to
decreased chlorophyll-a as nutrients became limiting due to their isolation in the bottom waters and
reliable high frequency chlorophyll-a data are required to convincingly demonstrate this phenomenon.
Notwithstanding these uncertainties it may be the case that the temporary stratification, interspersed with
mixing events, observed here represents a 'sweet spot' providing both the resources, i.e. chlorophyll-a and
the labile DOC it produces, and optimal conditions (anoxia) for $CH_4$ production.

Predicting climate change effects on GHG emissions in a future warmer world is not straightforward, as
there are multiple interacting drivers which combine to shape the GHG emissions of lakes. However, this
study suggests that temporary stratification, which is increasingly recognised as prevalent in ponds and
shallow lakes (Holgerson et al., 2022) and is likely to become more common with continued climate
change impacts (Woolway and Merchant, 2019) is likely to increase GHG emissions. This will be
particularly the case in more eutrophic systems where abundance algal derived dissolved organic matter
can fuel $CH_4$ production (Zhou et al., 2019).

The combination of high frequency data on water temperature and dissolved oxygen combined with
weekly measurements of GHGs increase the reliability of the findings presented here. Up until relatively
recently it has been assumed that for shallow lakes, such as Ormstrup lake, stratification is not an
important feature. Sampling has therefore focused on the surface layers of water bodies, using dissolved
concentrations of gases or floating chambers to characterise flux, e.g. (Davidson et al., 2015; Audet et al.,
2020; Peacock et al., 2021). Thus, most studies have overlooked bottom waters and do not have the
temporal resolution required to capture turnover flux emissions from surface measurements. Furthermore,
whilst many studies now include estimates of bubble emissions of $CH_4$ e.g. (Bergen et al., 2019), the
necessary temporal resolution to accurately characterise ebullitive emission is not well established. The
finding here indicated that in such dynamic systems near continuous measurement is desirable and that
short term collection over one or two days could provide massive, over or underestimate of $CH_4$
ebullition.

Our results show very large temporal variation in emissions of all three gases, but in particular $CO_2$ and
$CH_4$, and this highlights the need for high frequency measurements to accurately characterize emissions
from lakes. Even the weekly frequency of the sampling in this study was not sufficient to directly measure
all the emission pathways and turnover flux had to be inferred from bottom water calculations. These data
show that to capture the extent of GHG emissions from lakes it is vital we include all forms of flux,
including ebullition and turnover flux. Recent work has highlighted the fact that most emissions of $CH_4$
(over 50%) from fresh waters come from highly variable systems (Rosentreter et al., 2021), with the mean
and median emission rates of $CH_4$ differing greatly, indicating a few large emitters are responsible for a
large proportion of emissions. The sampling frequency applied here is rare, if a more standard resolution
of monthly measurements was applied the emissions estimate of all the gases, but in particular $CH_4$,
would be highly dependent on what phase of the stratification was captured. As an example, a monthly
sampling frequency could potentially miss all the stratification peaks - consequently massively
underestimating emissions, whereas a different sampling frequency could catch a number of peaks and
give a much higher estimate. Thus, the same sampling frequency on the same lake, but timed differently
could lead to conclusions of highly variable emissions. Consequently, in these highly dynamic systems
where temporary stratification occur in summer, high frequency measurements are required to accurately
estimate emissions. This is possible through eddy covariance approaches capable of capturing short term
changes and covering a large area (Erkkilä et al., 2018) but the cost of these systems means they are not
scalable to many sites. An increasingly accessible alternative is the use of automatic flushing chambers
using low cost sensors (Bastviken et al., 2020), which provide the potential for affordable high spatial and
temporal resolution measurement of GHG dynamics. This is a requisite for understanding the drivers of
GHG dynamic, which is required for being able to predict how they will respond in a range of scenarios
related to land use, climate change and management interventions.

**Code/data availability**
The datasets generated during and/or analysed during the current study are not publicly available as they
form part of ongoing research projects but are available from the corresponding author on reasonable
request and will be made publicly available later in the research project.

**Author contributions**
MS secured the funding for the wider lake restoration research project supplying the data. TAD, MS and
JA conceptualized the gas study. TAD and AN established the buoy and sensor system. EL, CE, TAD,
TB and JA collected and analysed the data. TAD wrote the paper and all authors commented on earlier
versions and read and approved the final draft.
**Competing interests**
The authors declare that they have no conflicts of interest.

## Acknowledgement

We thank our splendid technician team of Lene Vigh, Malene Kragh, Dorte Nedergaard and Dennis Hansen for their extreme competence on the lab and the field. We acknowledge Theis Kragh for the depth map of the lake already published in Søndergaard et al. 2023. We would also like to acknowledge the hard and diligent work of two anonymous reviewers and the associate editor David McLagan who very significantly improved the quality of the paper.  We are very grateful to the Poul Due Jensen Foundation for providing great support for this work and the Ormstrup project generally. TAD and CE were also supported by GREENLAKES (No. 9040-00195B) and The European Union's Horizon 2020 research and innovation programmes under grant agreement No 869296—The PONDERFUL Project.

Table 1. Summary lake information, summer mean values and (standard deviation) of a range of
variables

| Variable | n | Year 2020 |
|---|---|---|
| Secchi depth (m) | 22 | 0.86 (0.28) |
| Chlorophyll a ($\mu$g/l) | 20 | 53.4 (28.9) |
| pH | 22 | 8.04 (0.77) |
| Total phosphorus (mg/l) | 22 | 0.58 (0.11) |
| Total nitrogen (mg/l) | 22 | 1.50 (0.41) |


Table 2. Mean greenhouse gas flux (units $CO_2$: mg $CO_2$-C m$^{-2}$ day$^{-1}$, $N_2O$: mg $N_2O$ -N m$^{-2}$ day$^{-1}$, $CH_4$ both
diffusive and ebullitive in mg $CH_4$-C m$^{-2}$ day$^{-1}$) from the lake from spring to Autumn 2020. The emissions
are divided in diffusive, ebullitive emissions. The mean values for all the surface water stations and all
four transects of chambers are given. Emissions area separated into mixed versus stratified phases and
there SD are also given.  Ebullition was collected for a period covering two weeks so on a number of
occasion covered both mixed and stratified periods thus ebullition has a third category where both
mixed and stratified conditions occurred is given. Ebullition was significantly different across the three
phases, diffusive fluxes were not significantly different for $p$ values of 0.05.



| Emission type | gas | mean | mixed | Stratified | Strat and mixed |
|---|---|---|---|---|---|
| Diffusive | $CO_2$ | 493.7 (*529.6*) | 559.6 (*433.1*) | 449.8 (*587.6*) | |
| | $CH_4$ | 9.47 (*16.0*) | 5.9 (*4.1*) | 12.7 (*20.2*) | |
| | $N_2O$ | 0.11 (*0.09*) | 0.09 (*0.08*) | 0.12 (*0.11*) | |
| Ebullition | $CH_4$ | 17.28 (*19.62*) | **4.84** (***3.44***) | **47.29** (***21.95***) | **12.74** (***10.34***) |



Figure legends
Figure 1. Ormstrup lake bathymetry and sampling stations for surface water greenhouse gas sampling
(St1, St2, St3) bottom waters were sampled at S3. Transects of 10 bubble traps were placed on T1- T4.
Adapted from the Søndergaard et al. 2023.
Figure 2 Temperature profile from June 2020 when the buoy was deployed and surface and bottom water
oxygen from June to the end of September 2020. Manual chlorophyll-a ($\mu$g L$^{-1}$) values are also given in
the top panel.
Figure 3. Dissolved $CH_4$ concentrations from surface and bottom waters – thermal stratification periods
highlighted in grey and the white background indicate mixed waters
Figure 4 . Dissolved $CO_2$ concentrations from surface and bottom waters–thermal stratification periods
highlighted in grey and the white background indicate mixed waters
Figure 5 Dissolved $N_2O$ gas concentrations surface and bottom  thermal stratification periods highlighted
in grey and the white background indicate mixed waters
Figure 6. Omstrup lake surface fluxes of the $CH_4$, $CO_2$ and $N_2O$ gases based on dissolved conentration ,
thermal stratification periods highlighted in grey and the white background indicate mixed waters
Figure 7. Plot of $CH_4$ ebullition averaged for each transect (10 chambers per transect), data collected from
40 traps every two weeks. Thermal stratification periods highlighted in grey and the white background
indicate mixed waters.
Figure 8 – Total lake emissions per gas over the growing season in $CO_2$ equivalents. The emissions are
divided different emission modes: Diffusive, ebullitive and turnover flux.  All estimates constain some
uncertainty, in particular ebulltive flux is an underestimate and the turnover flux also contains a gret deal
of uncertainty.
Figure 9. Summary of the quantities of the gases present in the water and the volumes emited from the
different pathways. The size of the arrow is proportional to the emissions from each pathway and with the
startified state on the left and the mixed state on the right, with the turnover flux in the centre.

Figures and legends

Figure 1.

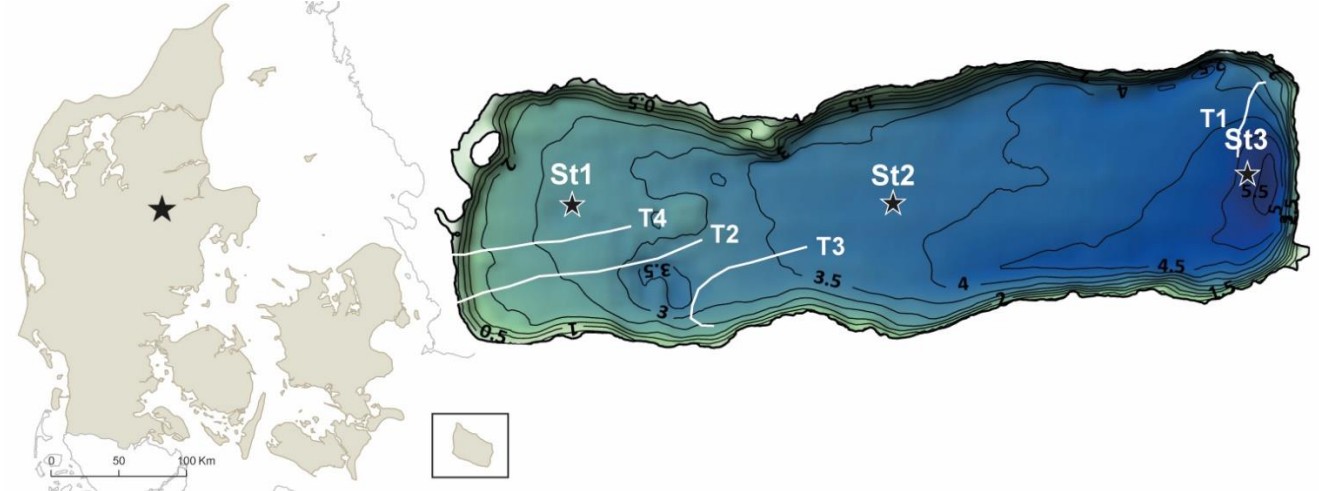


Figure 1. Ormstrup lake bathymetry and sampling stations for surface water greenhouse gas sampling (S1,
S2, S3) bottom waters were sampled at S3. Transects of 10 bubble traps were placed on T1- T4. Adapted
from the Søndergaard et al. 2023.

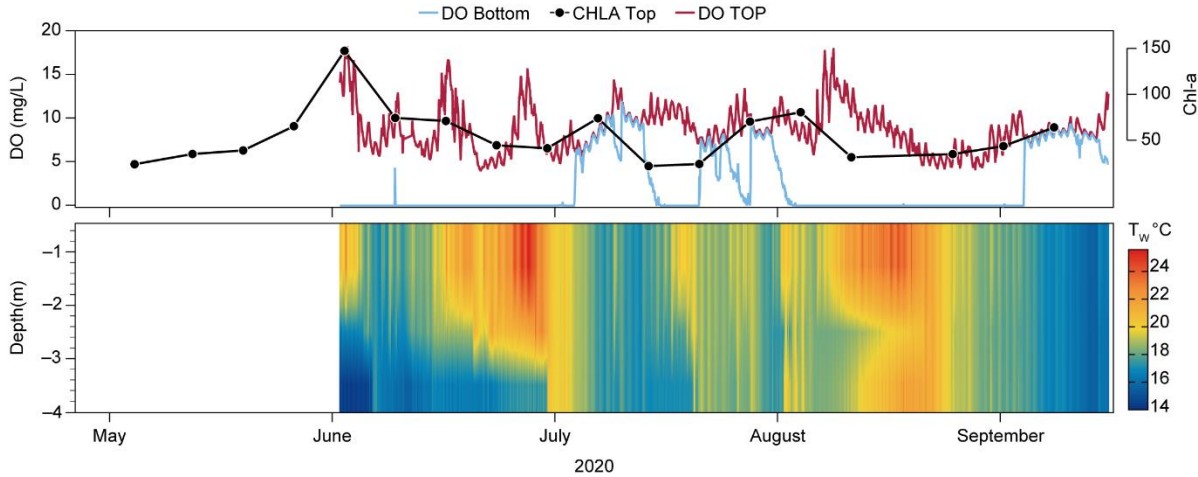


Figure 2 Temperature profile from June when the buoy was deployed and surface and bottom water
oxygen from June to the end of September. Chlorophyll-a ($\mu$g L$^{-1}$) values are also given in the top panel
and surface (DO TOP) and bottom (DO Bottom) dissolved oxygen (mg L$^{-1}$) are also given

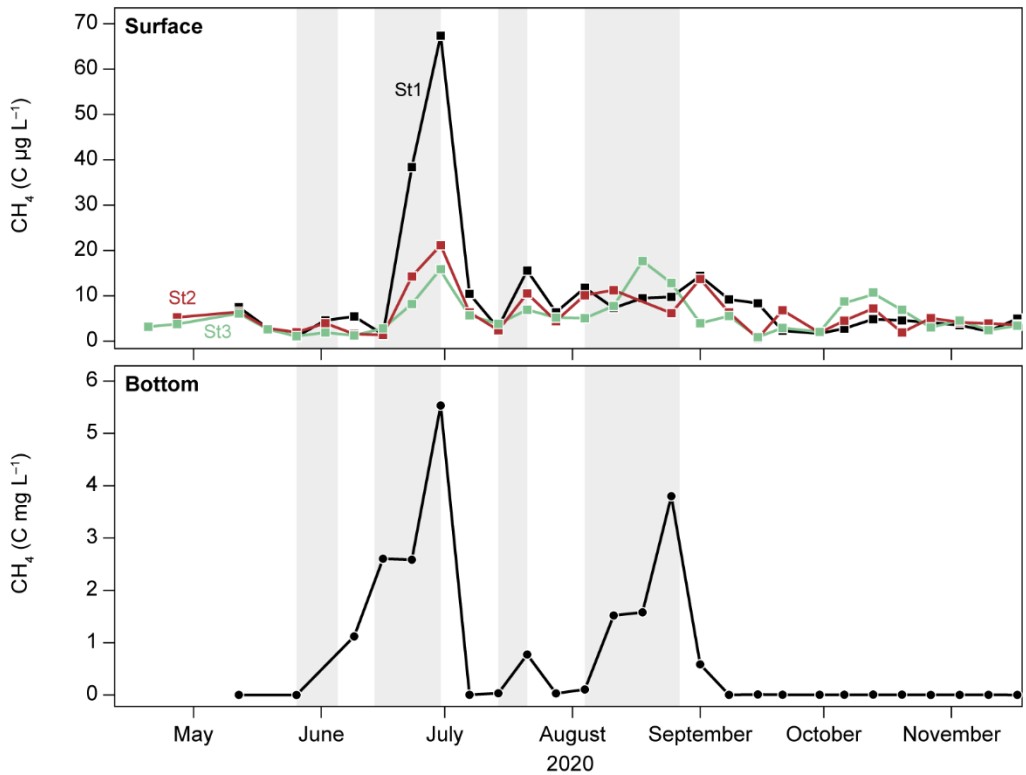


Figure 3. Dissolved CH$_4$ concentrations from surface and bottom waters – thermal stratification periods
highlighted in grey; white background indicate mixed waters. Note different y axis scales

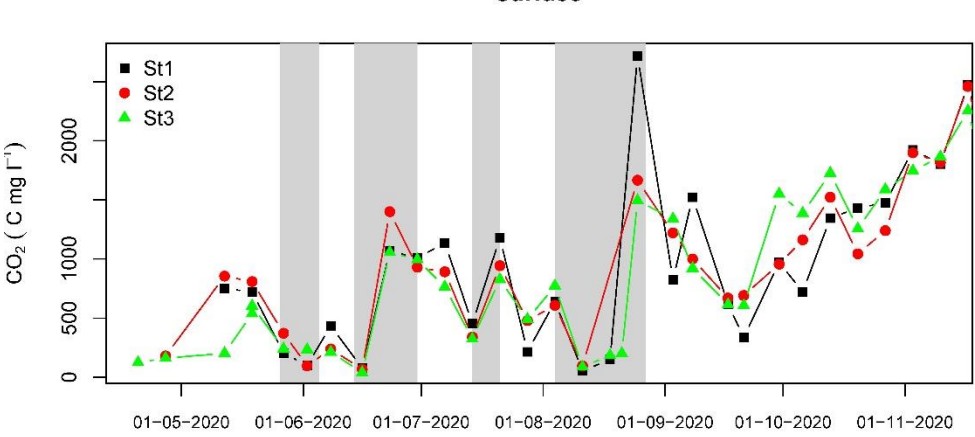

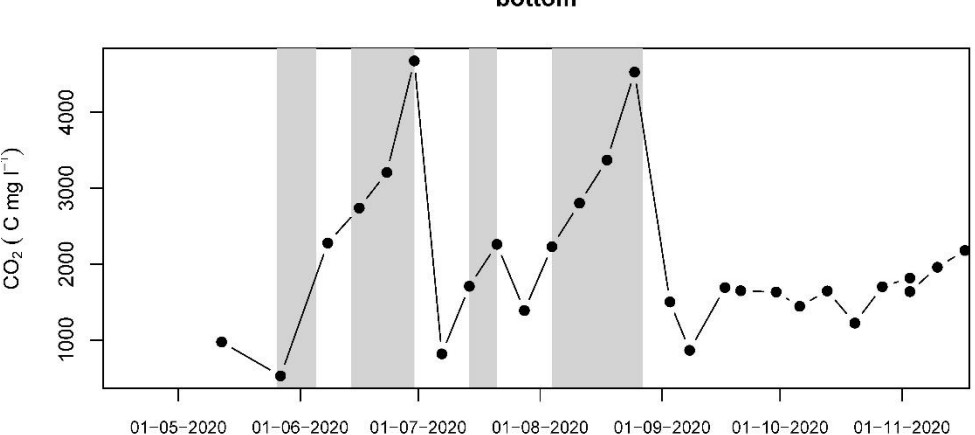


Figure 4 . Dissolved $CO_2$ concentrations from surface and bottom waters–
thermal stratification periods highlighted in grey; white background indicate mixed waters

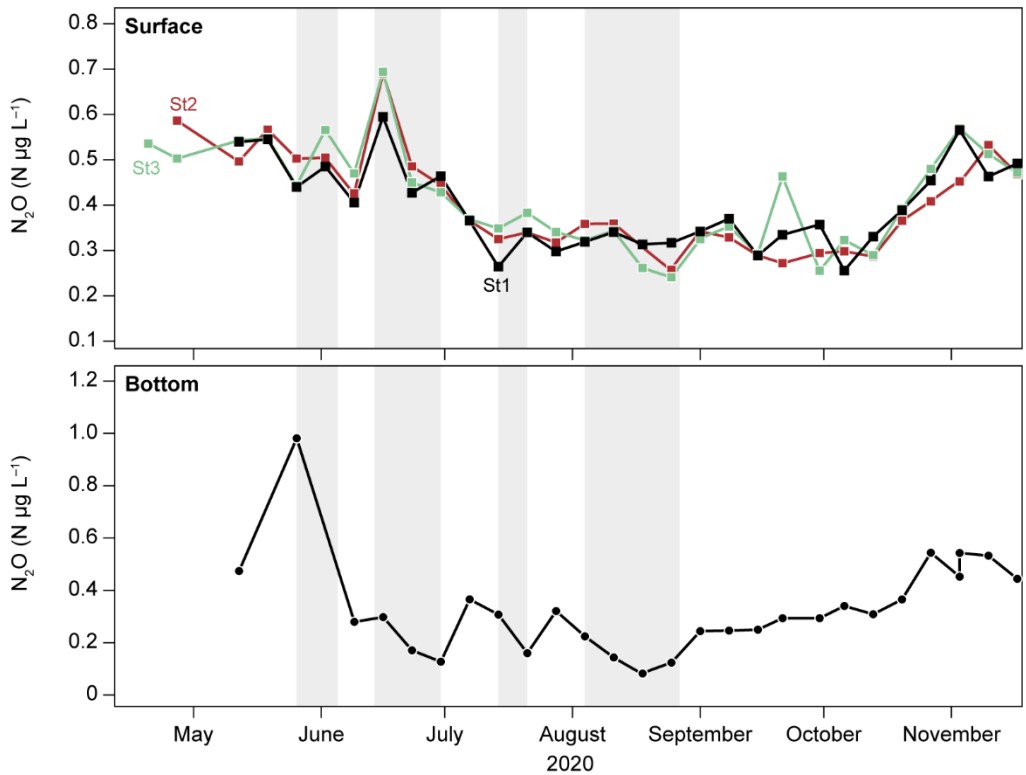


Figure 5 Dissolved N$_2$O gas concentrations surface and bottom  thermal stratification periods highlighted
in grey; white background indicate mixed waters

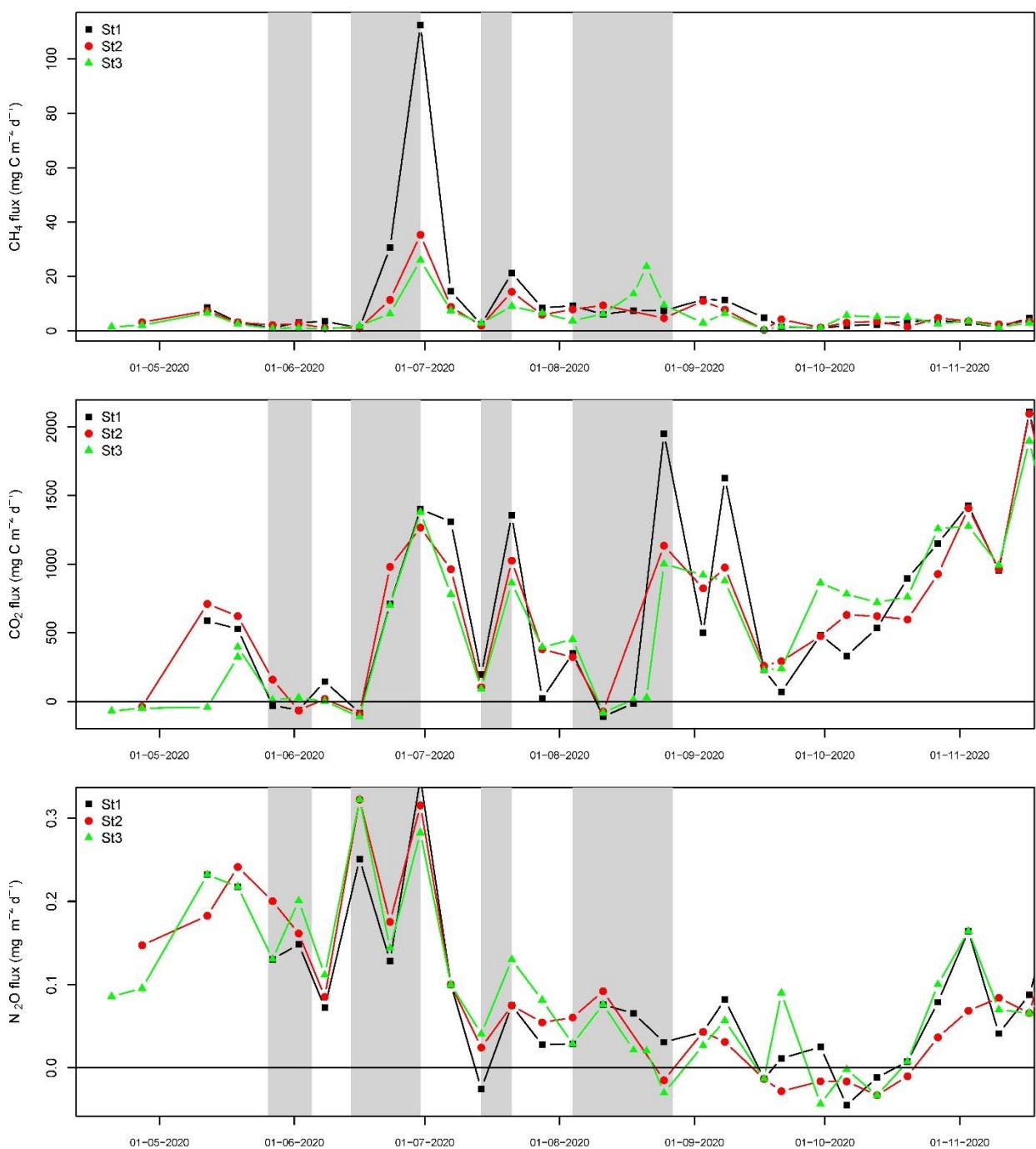


Figure 6. Omstrup lakesurface fluxes of the $CH_4$, $CO_2$ and $N_2O$ gases based on dissolved conentration ,
thermal stratification periods highlighted in grey; white background indicate mixed waters

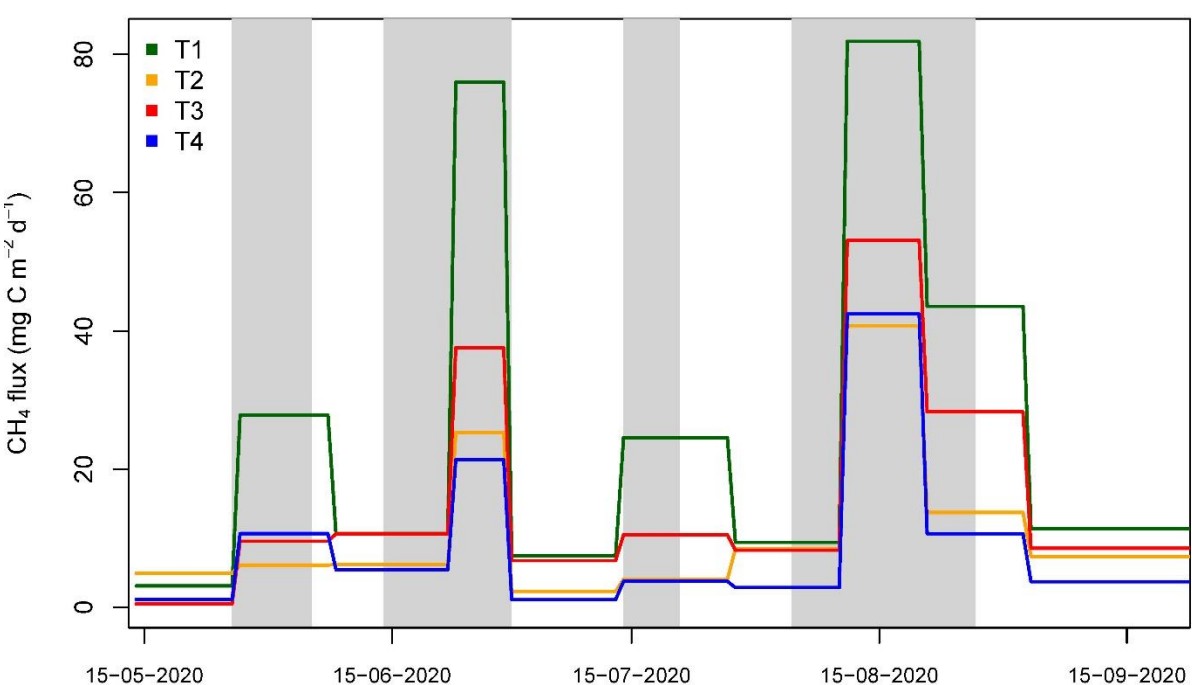


Figure 7. Plot of CH$_4$ ebullition averaged for each transect (10 chambers per transect), data collected from
40 traps every two weeks. thermal stratification periods highlighted in grey; whitle background indicate
mixed waters.

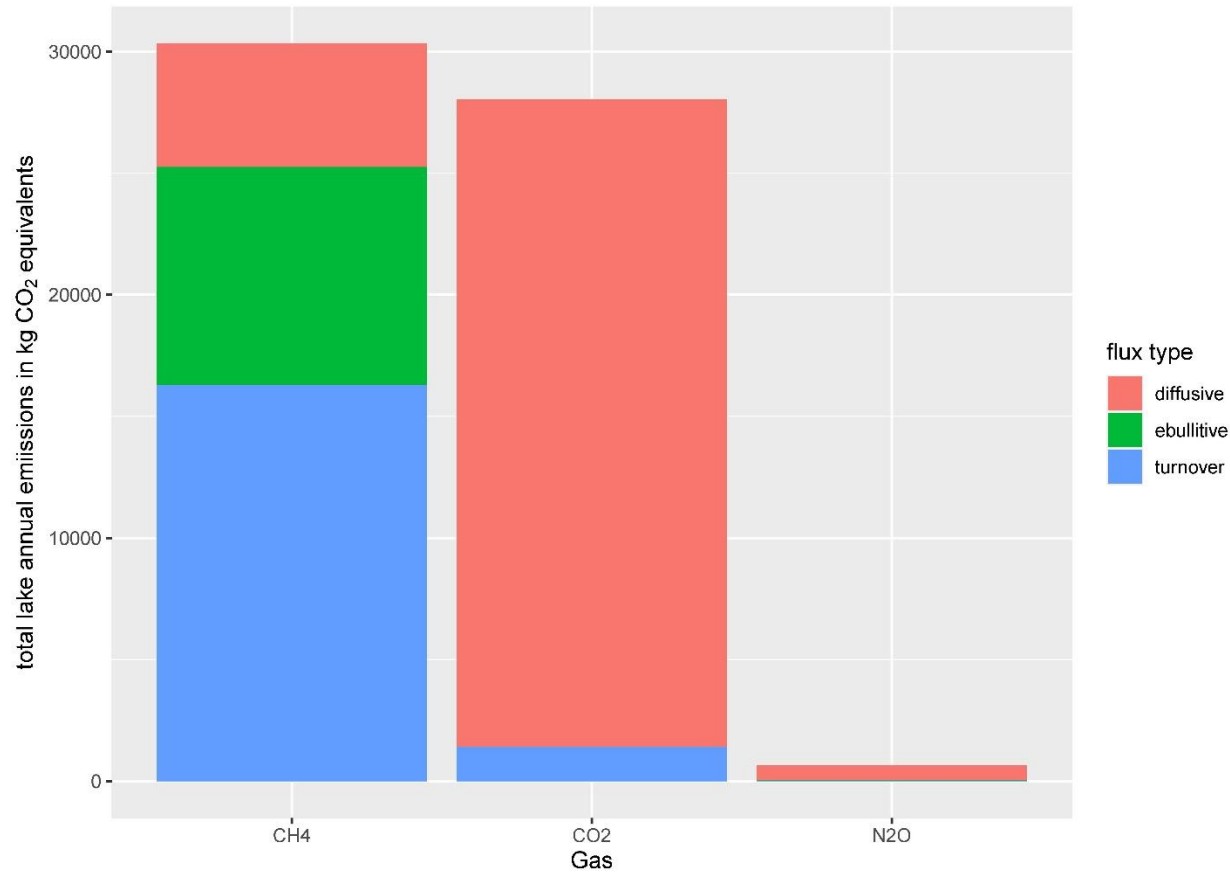


Figure 8 – Total lake emissions per gas over the growing seasson in $CO_2$ equivalents. The emissions are

divided different emission pathways: Diffusive, ebullitive and turnover flux.



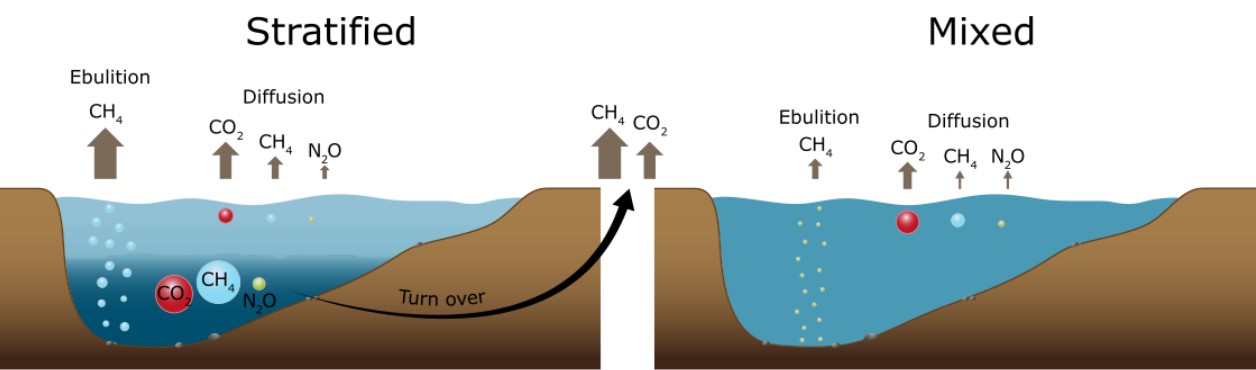


Figure 9  Summary of different flux types (bubble, diffusive and turnover) for the main greenhouse gases

($CH_4$ $CO_2$ and $N_2O$) observed between the stratified and mixed phases at Ormstrup lake patterns in the

stratified and mixed phase. The turnover flux of $CH_4$ and $CO_2$ is also represented. The size of the arrow

represents the relative amount of emission and the size of the circle in the lake represents the
concentration of dissolved gases in stratified or mixed water column.

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
