# Peer review of "Temporary stratification promotes large greenhouse gas emissions in a shallow eutrophic lake Thomas A Davidson1, 2, Martin Søndergaard1, 2, 3, Joachim Audet1, 2, Eti Levi1, Chiara Esposito1, 2, Tuba Bucak Onay1, Anders Nielsen1,4"

_Biogeosciences, 2023_

## Referee Comment (RC1)

**Reviewer comments for Preprint bg-2023-43 'Temporary stratification promotes large greenhouse gas emissions in a shallow eutrophic lake'**

*Overall comments*

In this study, greenhouse gas (GHG) samples were taken over a 6-month period from a shallow lake in Denmark with the aim of understanding stratification and mixing effects on GHG fluxes. The paper provides an interesting data set, with surface and bottom water GHG concentrations resolved in addition to ebullitive fluxes. The identification of turnover as a highly transient event that can contribute significantly to lake GHG budgets is an important finding.

The paper is well written, the figures are clear and the discussion provides a succinct description of the findings. I have two main points for the author to review, and also provide some minor editorial comments.

The key question of this study was to understand how ebullitive and diffusive fluxes of the key GHGs: $CH_4$, $CO_2$ and $N_2O$ respond to temporary thermal stratification. However, $N_2O$ is not mentioned in the discussion in this paper and I therefore do not feel that the question has been adequately addressed. How important was $N_2O$ in the overall lake budgets, and were accompanying nutrient data able to help understand nitrification/denitrification pathways that might result in flux changes through stratification and mixing? I also felt that the discussion was heavily weighted towards $CH_4$ though the key question concerns all major GHGs. I would have expected that $CO_2$ undersaturation might have been detected via headspace sampling at times of high algal productivity, as has been observed in shallow lakes in the tropics (e.g. Borges et al. 2022) and that this would mean the lake is a $CO_2$ sink at some points. That this did not occur is of interest. It is also important to note that in lakes with pH > 7.5 there is a need to further correct headspace derived $CO_2$ data, as outlined in Koschorreck et al. 2021, to account for chemical equilibration of the carbonate system.

The discussion does not place the fluxes reported in this study in a wider context with the exception of a brief comparison of mean $CH_4$ fluxes to a global study by Rosenterer et al. (2021). I therefore found it difficult to understand how important or significant the fluxes were from this lake. I think there should be more explicit comparison across all three GHGs with comparative studies from both equivalent climate zones and in a global context.

*Minor comments*

| | |
|---|---|
| Line 27-28 | Missing 'for' – change to "also the need for high frequency measurements of GHG emission in 28 order to accurately characterise emissions from temporarily stratifying lakes." |
| Line 35 | Should this be 'Freshwaters'? |
| Line 55 | 'Identity' is a bit odd in this context |
| Line 72-74 | Add reference here |
| Line 119 | Sampling duration not clear. State start and end point of sampling. |
| Line 120 | Measurement according to 'Danish standard procedures' doesn't mean much for international readers. A brief additional explanation would be beneficial. |
| Line 128 | Did water level changes influence the relative distance between the surface and bottom water sampling points over the sampling duration? |
| Lines 191-203 | This seems like material for discussion rather than methods as it critiques the method applied rather than describes it objectively. |
| Lines 212-213 | The GWPs cited come from two separate IPCC reports. The latest report, AR5 (though AR6 is due imminently…), states the 100 yr time horizon GWPs for |

|  |  |
|---|---|
|  | methane and nitrous oxide as 28 and 265 respectively. Suggest using these for consistency. |
| Line 230 | Remove interpretation from results 'likely limited by nitrogen (Søndergaard et al., 2023)' |
| Line 231 | Change to 'mixing event' |
| Line 240 | Change to 'coincident' |
| Line 294 | Remove 'massive' |
| Line 302 | Remove more from 'more lake-wide driver' |
| Line 306-307 | This sentence is confusing. Do you mean: '6 Thus, whilst we do not have direct evidence it seems more likely that these increased emissions in the littoral zone were driven at least in part by the partial, wind-driven mixing of the GHG rich bottom waters.'? |
| Line 318 | The Wik et al (2013) study was focussed on Arctic lakes and found a seasonal shift in contribution of ebullition to total methane flux whereby the dominance of shallow zone bubble $CH_4$ fluxes decreased over summer relative to an increase in intermediate and deep zone fluxes. This suggests a strong temperature control. Perhaps a caveat could be added to this comparison for clarity. |
| Lines 327-335 | Agree, and important to state, but equally weekly headspace sampling has some of the same issues whereby GHG fluxes resulting from highly dynamic mixing/stratification processes may not be adequately resolved.

I see this caveat has been added later in the discussion (lines 400-401).

I suggest adding in that eddy covariance flux measurements are a way to achieve high temporal resolution data to characterise these processes, including the turnover flux that is described as occurring over just a few hours (e.g. Erkkilä et al. 2018; Podgrajsek et al. 2014). |
| Lines 382-384 | Nutrient enriched sediments would likely provide a stable source of organic matter as redox conditions promote internal loading from sediments. |
| Table 1 | Add standard deviations and how many observations (n) informed the mean. |
| Figure 1 | Where is the Aqua troll located? |
| Figures 3-5 | Suggest merging into one figure with multiple panels |
| Figure 7 | I am not sure this works as a line plot. Perhaps just plot the data as points, otherwise huge step changes in ebullition fluxes are implied. |

**References**

Alberto V. Borges, Loris Deirmendjian, Steven Bouillon, William Okello, Thibault Lambert, Fleur A.E. Roland, Vao F. Razanamahandry, Ny Riavo G. Voarintsoa, François Darchambeau, Ismael A. Kimirei, Jean-Pierre Descy, George H. Allen, Cédric Morana Greenhouse gas emissions from african lakes are no longer a blind spot Sci. Adv., 8 (25) (2022), Article eabi8716, 10.1126/sciadv.abi8716.

Erkkilä, K.-M., Ojala, A., Bastviken, D., Biermann, T., Heiskanen, J. J., Lindroth, A., Peltola, O., Rantakari, M., Vesala, T., and Mammarella, I.: Methane and carbon dioxide fluxes over a lake: comparison between eddy covariance, floating chambers and boundary layer method, Biogeosciences, 15, 429–445, https://doi.org/10.5194/bg-15-429-2018, 2018.

Koschorreck, M., Prairie, Y. T., Kim, J., and Marcé, R.: Technical note: $CO_2$ is not like $CH_4$ – limits of and corrections to the headspace method to analyse $pCO_2$ in fresh water, Biogeosciences, 18, 1619–1627, https://doi.org/10.5194/bg-18-1619-2021, 2021.

Podgrajsek, E., Sahlée, E., Bastviken, D., Holst, J., Lindroth, A., Tranvik, L., and Rutgersson, A.: Comparison of floating chamber and eddy covariance measurements of lake greenhouse gas fluxes, Biogeosciences, 11, 4225–4233, https://doi.org/10.5194/bg-11-4225-2014, 2014.

---

## Author Comment (AC1)

**Reviewer 1 comments for Preprint bg-2023-43 'Temporary stratification promotes large greenhouse gas emissions in a shallow eutrophic lake'**

Reply: General comments : we are grateful to the reviewers for their thoughtful comments and general enthusiasm for the work. But also for more critical comments which we think have improved the quality of the work substantially. We have addressed each point in turn in the following document.

*Overall comments*
In this study, greenhouse gas (GHG) samples were taken over a 6-month period from a shallow lake in Denmark with the aim of understanding stratification and mixing effects on GHG fluxes. The paper provides an interesting data set, with surface and bottom water GHG concentrations resolved in addition to ebullitive fluxes. The identification of turnover as a highly transient event that can contribute significantly to lake GHG budgets is an important finding.
The paper is well written, the figures are clear and the discussion provides a succinct description of the findings. I have two main points for the author to review, and also provide some minor editorial comments.

The key question of this study was to understand how ebullitive and diffusive fluxes of the key GHGs: $CH_4$, $CO_2$ and $N_2O$ respond to temporary thermal stratification. However, $N_2O$ is not mentioned in the discussion in this paper and I therefore do not feel that the question has been adequately addressed. How important was $N_2O$ in the overall lake budgets, and were accompanying nutrient data able to help understand nitrification/denitrification pathways that might result in flux changes through stratification and mixing?

Reply : Some text on N2O was added to the discussion clarifying the very small role it has in GHG dynamics in the lake and that the emissions patterns are not strongly related to the stratification.

I also felt that the discussion was heavily weighted towards $CH_4$ though the key question concerns all major GHGs. I would have expected that $CO_2$ undersaturation might have been detected via headspace sampling at times of high algal productivity, as has been observed in shallow lakes in the tropics (e.g. Borges et al. 2022) and that this would mean the lake is a $CO_2$ sink at some points. That this did not occur is of interest.

Reply: The CO2 dynamics are discussed a little more and periods of influx highlighted.

It is also important to note that in lakes with pH > 7.5 there is a need to further correct headspace derived $CO_2$ data, as outlined in Koschorreck et al. 2021, to account for chemical equilibration of the carbonate system.

Reply: The Koschorreck et al. 2021 correction was applied to the data and there were periods where the % error in estimation of dissolved concentration was large (more than 40%). The absolute difference was not very significant, but there was a small increase the periods of influx a little and the relevant figures (4, 6 and 8 ) have been changed.

The discussion does not place the fluxes reported in this study in a wider context with the exception of a brief comparison of mean $CH_4$ fluxes to a global study by Rosenterer et al. (2021). I therefore found it difficult to understand how important or significant the fluxes were from this lake. I think there should be more explicit comparison across all three GHGs with comparative studies from both equivalent climate zones and in a global context.

The fluxes are now placed in a wider context by comparison with other work from similar climates.

| | |
|---|---|
| *Minor comments* Line 27-28 | Missing 'for' – change to "also the need for high frequency measurements of GHG emission in 28 order to accurately characterise emissions from temporarily stratifying lakes."
Reply: done |
| Line 35 | Should this be 'Freshwaters'?
Reply: Fresh waters is two words as a noun, one as an adjective |
| Line 55 | 'Identity' is a bit odd in this context
Sentence changed |
| Line 72-74 | Add reference here
Reply: done |
| Line 119 | Sampling duration not clear. State start and end point of sampling.

Reply: It is a bit confusing, so I have clarified each section |
| Line 120 | Measurement according to 'Danish standard procedures' doesn't mean much for international readers. A brief additional explanation would be beneficial.
Reply: done |
| Line 128 | Did water level changes influence the relative distance between the surface and bottom water sampling points over the sampling duration?
Reply: a little but the relative distance from the lake bed of the bottom samples was consistent. When water levels were lower the relative distance between top and bottom samples would have been less, but |

water level did not change so much as it was wet summer.

| | |
|---|---|
| Lines 191-203 | This seems like material for discussion rather than methods as it critiques the method applied rather than describes it objectively. I have expanded this section and would be happy to place it in the discussion or even as a supplement. |
| Lines 212-213 | The GWPs cited come from two separate IPCC reports. The latest report, AR5 (though AR6 is due imminently…), states the 100 yr time horizon GWPs for |

methane and nitrous oxide as 28 and 265 respectively. Suggest using these for consistency.
Done, thanks.

| | |
|---|---|
| Line 230 | Remove interpretation from results 'likely limited by nitrogen (Søndergaard et al., 2023)' ok |
| Line 231 | Change to 'mixing event' done |
| Line 240 | Change to 'coincident' done |
| Line 294 | Remove 'massive' done |
| Line 302 | Remove more from 'more lake-wide driver' done |
| Line 306-307 | This sentence is confusing. Do you mean: '6 Thus, whilst we do not have direct evidence it seems more likely that these increased emissions in the littoral zone were driven at least in part by the partial, wind-driven mixing of the GHG rich bottom waters.'? Yes thanks! |
| Line 318 | The Wik et al (2013) study was focussed on Arctic lakes and found a seasonal shift in contribution of ebullition to total methane flux whereby the dominance of shallow zone bubble $CH_4$ fluxes decreased over summer relative to an increase in intermediate and deep zone fluxes. This suggests a strong temperature control. Perhaps a caveat could be added to this comparison for clarity. |

done

| | |
|---|---|
| Lines 327-335 | Agree, and important to state, but equally weekly headspace sampling has some of the same issues whereby GHG fluxes resulting from highly dynamic mixing/stratification processes may not be adequately resolved. I see this caveat has been added later in the discussion (lines 400-401). |
| | I suggest adding in that eddy covariance flux measurements are a way to achieve high temporal resolution data to characterise these processes, including the turnover flux that is described as occurring over just a few hours (e.g. Erkkilä et al. 2018; Podgrajsek et al. 2014). done |
| Lines 382-384 | Nutrient enriched sediments would likely provide a stable source of organic matter as redox conditions promote internal loading from sediments. agree |
| Table 1 | Add standard deviations and how many observations (n) informed the mean. done |
| Figure 1 | Where is the Aqua troll located? 0.5 m depth |
| Figures 3-5 | Suggest merging into one figure with multiple panels Can do – we leave them as 3 now and if accepted the editor can decide which is better. |
| Figure 7 | I am not sure this works as a line plot. Perhaps just plot the data as points, otherwise huge step changes in ebullition fluxes are implied. We thought about this a lot and had it as points before, but the data are the mean ebullition of the previous 14 days – so the line plot is the most accurate way of presenting it. |

---

## Author Comment (AC2)

**Reviewer 2 comments for Preprint bg-2023-43 'Temporary stratification promotes large greenhouse gas emissions in a shallow eutrophic lake'**

**Overall**

This work presents data on greenhouse gas concentrations and estimated fluxes, along with temperature, oxygen, and chlorophyll data, from a eutrophic shallow lake over one season (April / May – October 2020). The topic is timely as the scientific community is working to reduce uncertainty around aquatic greenhouse gas emission estimates, especially from shallow systems. While I found the research interesting, I have a few overarching concerns:

1. The greenhouse gas data have been previously published (Søndergaard et al. 2023), which already describes the novel results of this paper: that temporary stratification events lead to the buildup of greenhouse gases, which are likely released upon turnover. I think the authors can more clearly describe how this work is different from the previously published paper-- I suspect the new addition is that this paper estimates the ebullitive and turnover fluxes, which I address next.

Re. A limited amount of data (some weeks of concentration data) was published in the Søndergaard et al. 2023, which is a summary of the different types of lake ecosystem responses to stratification – covering fish behavior, nutrient dynamics, algal biomass. The current MS is focused in on the GHG dynamics over a longer period and includes the estimates of different flux types. We think it is clear this this work is sufficiently different from that reported in Søndergaard et al. 2023 to stand alone.

2. Ebullitive fluxes were measured using floating chambers. I have not seen chambers used to estimate ebullitive fluxes before, unless the chambers were set for a short amount of time and concentrations were measured repeatedly (e.g., using a portable gas analyzer or manual sampling)—I've seen this done up to 24 hours. Then over that short amount of time, the diffusive and ebullitive fluxes are teased apart. In the current study, the chambers were deployed for 2 weeks at a time, during which the chambers would have equilibrated with the water, with the exception for bubbles. I do not think it is possible to determine the total amount of ebullition with this approach. For instance, if a bubble occurred on Day 1, the $CH_4$ could diffuse back into the water column by the time the chambers were checked two weeks later. While I am empathetic to the challenges of measuring ebullition, I do not agree with the authors that this is the "least worst method available."

Reply: This is a very relevant point and something we have given a lot of thought. We have added a section entitled to the methods discussion the static chambers to estimate ebulltion. This can be moved to supplementary materials as a stand alone section 'on the use of static chambers to estimate ebulltion' if it is more appropriate. As the reviewer states the method is imperfect in two main ways

1. there is constant diffusion of CH4 into the chamber to an equilibrium value which will be reached over the time of deployment:
2. We do not know when the bubbles arrive in the chamber and in the worse case if a bubble arrives on day 1 then there is time for the high concentrations in the bubble to diffuse back into the water.

We have attempted to tackle this uncertainty in two ways: 1) To establish evidence of ebullition we used the dissolved concentration of the $CH_4$ on the day sampling and calculated the theoretical concentration of $CH_4$ in the chamber at equilibrium with the dissolved concentration. If the $CH_4$ of ppm in the chamber was lower or close (<10 ppm) to the theoretical values we ascribe an ebullition a value of zero to the chamber. If the value was higher this is evidence that ebullition had occurred over the two week period. There is then a choice as to whether: 1) use this corrected value to estimate ebullition, or 2) use the value measured in the chamber without correction. We chose to not correct these values. The rationale being that once ebullition has occurred the $CH_4$ concentration in the chambers is higher than in the water and so any diffusion taking place will be from the chamber to the water. As the ebullition estimated by the static chambers is already highly likely to be an underestimate it does not seem wise to increase the underestimate by correcting for the diffusive flux.

In addition, we took new measurements from static chambers as done for this study and compared the results with an automatic flushing chamber (AFC) which was placed alongside the static chamber. The AFC is an adaptation of the chambers developed by Bastviken et al ((Bastviken et al., 2020; Duc et al., 2013) where the same low cost sensors as applied by Bastviken et al. 2020 (Figaro TGS 261 E00) and an infra- red $CO_2$ (Sensiron SCD30) measuring at high frequency (every few seconds). The chambers have a fan based flushing system which ran every four hours. This allows ebullitive and diffusive flux to be calculated 6 times a day. The total flux is estimated from the concentration measured just prior to the chamber flushing after four hours of diffusive and ebullitive accumulation. In order to separate ebullition from diffusive flux we used an iterative process where 120 five minutes periods were randomly sampled over the four hours between flushes. Linear regression of change in $CH_4$ concentration against time was conducted on these 300 five minute periods and the median beta of the regression used as the estimate of diffusive flux. This iterative process obviates the effects of any bubbles which can arrive in these five minute periods used to estimate diffusive flux. A bubble arriving in the chamber in these five minutes period would result in a low $r^2$ of the resultant regression. Tests show that the $r^2$ and beta of the regression stabilize after around 120 iterations but 150 was chosen so to ensure that even in periods of frequent ebullition the

diffusive flux could be reliably estimated. Once diffusive flux was reliably estimated for each period then it was possible to disentangle diffusive and ebullitive flux for each four hour period.

The comparison of the two methods took place in June and July 2023 the results of the AFC gave daily estimates of $CH_4$ ebullition from which an average estimate of ebulltion for the sampling period was derived. These averages were for the sampling period ending on the 10/07/2023 were 33.3 mg $CH_4$- C $m^{-2}d^{-1}$ and for the period ending 18/07/2023 was 91.2 $CH_4$- C $m^{-2}d^{-1}$. The estimate from the static chamber which the AFC was placed next to was 27.9 mg $CH_4$- C $m^{-2}d^{-1}$ and 59.9 mg $CH_4$- C $m^{-2}d^{-1}$ which were underestimates of 5.4 and 31.3 mg $CH_4$- C $m^{-2}d^{-1}$, which represents a 16 and 31 % underestimate.

31% is a relatively large error and this may have been higher in the second testing phase as the highest ebullitive fluxes arrived on the first day of the period. Therefore 30% may be a maximum error. When these results are compared with the error in the estimate of using a single day's observation (supp material 2 and 3) to characterize ebullitive flux which had an error of 4-111% error and a median error of 50% even an error of 31% is relatively low. In summary we can be sure that static chambers are an underestimate of ebullition but they provide a feasible means of continuous data collection which the results show have greater merit than the deployment of 'better' methods for shorter periods of time.

3. Turnover fluxes assume that all the CO2 and CH4 gases in the hypolimnion were released when the lake mixed. This approach assumes that there was no CH4 oxidation during turnover, which contrasts previous studies (see Kankaala et al. 2007, Thottathil et al. 2019, Zimmerman et al. 2021). If the lake mixes rapidly, oxidation may be low; however, previously published data on this lake from the same year (Søndergaard et al. 2023) shows that while thermal mixing can occur within hours, it can take 5 days for the complete mixing of oxygen. As thermoclines and oxyclines are offset in this lake (and this may be a common phenomenon, e.g. Gray et al. 2020), I don't think it's fair to assume oxidation is 0. Therefore, without oxidation estimates, these turnover values may be huge overestimates.

Another good point, our calculation of turnover flux is simplistic as we assumed that all the methane in the hypolimnion was released, whereas a portion of it would be oxidized. To address this we measured $CH_4$ oxidation rates in the surface waters of the lake and we have used these values to estimate the amount of $CH_4$ oxidised during turnover. Whilst the mixing of the lake started on 30[th] of June the bottom water was not oxic until four days later. We used a miniumum, mean and maximum oxidation rates measured to correct the estimate of overturn flux. Details are given the in the methods. The estimated overturn flux was reduced, but only by between 2 and 8%the estimate by the amount of $CH_4$ that would have been oxidized over the 4 days the lake took to fully mix. The mean $CH_4$ oxidation rates were used to correct the turnover flux estimates.

**Specific Comments**

Abstract

- Lines 18-19: provide details of length of the study (e.g., May to October for GHGs)

*done

Introduction

- Broad framework of the Introduction is focused on climate change, but this is not a climate change study. While climate change will likely change the mixing regimes of lakes and ponds, it is not the major focus of this study. The novelty of this study seems to be that intermittently mixing lakes have unique biogeochemical cycles, and the oxic-anoxic cycles may explain the variability in fluxes over time. The challenge is that this story is also the framework for the Søndergaard et al. 2023 paper, which used the same dataset.

  Reply: It is true this is not explicitly a climate change study, but as one of the effects of climate change will be more frequent heat waves and periods of lake stratification so we think that the introduction is appropriate. The intermittent mixing and its effects is indeed the focus and the comparison of the dynamics of the stratified versus the mixed phase is indeed the focus. A very small part of this dataset – concerning bottom and surface water concentrations of the GHGs before and after the mixing event at the end of June were in the Søndergaard et al 2023. We think that the current work is a standalone paper.

- Line 40: Provide more details on the % contribution coming from lakes and ponds instead of saying "large proportion"

  Reply: Altered to over half – as in the paper

- Line 53: See Deemer and Holgerson 2021 on the drivers of diffusive and ebullitive fluxes

  Reply: Ref added

- Lines 62-66: See Holgerson et al. 2022, which describes mixing regimes in shallow waterbodies including this category of intermittent or temporary stratification.

included

- Lines 71-73: The discussion of C burial is interesting but a bit of a red herring as it is not something addressed in this paper

  True we do not at all address it, it was included as it was part of the study cited here as such we keep it for now.

- Lines 76-77: How is this specific to shallow lakes?

  Good question! Have changed the way this part is written, I hope it is clearer

Methods

- Lines 90-98: Provide overview of the study time start and end in the first few paragraphs—the whole season study is a major strength of this study.

  done

- Line 99: Was a solar shield used for measuring air temperature?

  yes

- Lines 113-116: Stratification should be defined by density instead of temperature because the density-temperature relationship is not linear, and water density is what determines stratification. See Gray et al. 2020 for more details on the importance of using density over temperature. Especially considering this study includes measurements from May – October, the temperature range is large and the density-temperature relationship becomes more important

  The reviewer is correct that using density is better than temperature but here we define periods as either stratified or mixed on the stated criteria and we think this is sufficiently robust for this study.

- Lines 113-116: I find it confusing to determine stratification periods by both temperature and oxygen considering the thermocline and the oxycline set up at different time scales (e.g., Søndergaard et al. 2023). I recommend just using density differences.

  As above

- Lines 115-116: The statement that bottom waters remain undisturbed is an assumption—partial mixing events likely increase turbulence at the surface of the hypolimnion and gases can be exchanged.

  True – statement is caveated

- Lines 148-151: How far away was windspeed measured? Were the on-lake conditions compared to the institute's measurements?

  Wind data comes from the Danish meterological institute which provides modelled wind speed for 20km2 grid squares

- Lines 168-170: Why is oxygen used to determine the hypolimnion here, whereas mixing was previously defined based on temperature and oxygen?

  We used a the zero oxygen level to define the volume of the hypolimnion here based on the many profile data in order to calculate the volume of water that is likely to have the concentration of $CH_4$ measured in the bottom waters. Using the oxygen concentration provides a more conservative estimate than combining temp and oxygen

- Lines 170-171: As described above, I do not think the authors should assume oxidation is 0 when it may take 5 days for oxygen concentrations to equilibrate following isothermal conditions.

  See above

- Lines 177-178: Floating chambers need further description (surface area, volume).

  Done in methods

- Lines 177-188: See above concerns about estimating ebullition from chambers deployed for two weeks.'

  addressed

- Lines 201-203: I appreciate the caveats associated with the floating chambers, but as described above, I need more convincing that these methods are appropriate. Why not measure volume displaced and collect fresh bubbles? How do the methane concentrations in the chambers compare to fresh bubbles?

  There are a few other options and the floating chambers underestimate flux but we have provided evidence that they can provide useful information

Results

- Lines 218-219: Use more quantitative descriptions of time mixed vs. stratified. If you use the definition for mixed vs. stratified described in the methods (see critique on not using density), this will allow for quantifying mixed vs. stratified periods for broad summary.

  See previous comment

- Figure 2: I recommend using the same gray backgrounds to show periods of thermal mixing—this will help highlight the offset between isothermal conditions and oxygen.

  Coming if accepted

- Please use statistical tests and present the results on concentrations during mixed vs. stratified periods; e.g., lines 246-248.

  Done for fluxes in table 2-..

- Table 1: This appears to be right from Søndergaard et al. 2023. I recommend removing it and describing in the Methods, or providing standard error or standard deviation for the 2020 season. TN:TP would be more helpful in its molar ratio to examine nutrient limitation.

  Ok- SD and n provided as above

- Table 2: Statistical comparisons needed to compare mixed vs. stratified periods

  done

- Figure 3: Add where the bottom samples were taken from—Station 3?

  Done – yes as the deepest point –station 3

- Figure 9: Make sure the arrows match statistical differences observed.

Discussion

- Line 294: provide number instead of saying "massive"

  Removed – the other reviewer did not like it either

- Lines 295-297: provide statistical comparisons

As above

- Lines 306-308: The partial mixing here contrasts the assumption in the introduction that bottom waters were not affected by partial mixing events

  Earlier statement changed

- Lines 330-335: As described above, I still need convincing that the floating chamber method is appropriate to estimate ebullition.

  Already attempted

- Line 345: Explain why it's an overestimate (i.e., oxidation)

  It is corrected for now

- I think subheadings could help organize the discussion

- The conclusion doesn't tie back to the introduction framework focused on climate change, which again suggests that the Introduction should instead focus on variable mixing regimes in shallow lakes and the consequences for biogeochemical cycling.

Good point, we have added a sentence on the possible climate change effects

Minor Comments Not Requiring Response

- Line 27: add "for" between "need high"

  done

- Line 48: add comma after "approach"

  done

- Line 434: remove "crack" as in English, it often references cocaine, which I do not think is the intent here.

  It is more informal English for 'expert' and was an attempt at humour… but I can change it to avoid the drug connotations.

**Citation**: https://doi.org/10.5194/bg-2023-43-RC2

Bastviken, D., Nygren, J., Schenk, J., Parellada Massana, R., and Duc, N. T.: Technical note: Facilitating the use of low-cost methane (CH4) sensors in flux chambers – calibration, data processing, and an open-source make-it-yourself logger, Biogeosciences, 17, 3659-3667, https://doi.org/10.5194/bg-17-3659-2020, 2020.

Duc, N. T., Silverstein, S., Lundmark, L., Reyier, H., Crill, P., and Bastviken, D.: Automated flux chamber for investigating gas flux at water–air interfaces, Environ. Sci. Technol., 47, 968-975, 2013.

---

## Author Response (AR2)

Reviewer 1

The addition of the statement on eddy covariance as a means of assessing high temporal resolution GHG fluxes from lakes is jarring in its current position in the manuscript (lines 380-382). Suggest moving to the end of the following paragraph which highlights the need to measure across discrete, hard to predict events such as turnover.

**Moved to the final paragraph of the discussion**

Reviewer 2

I appreciate the edits the authors made on this manuscript, and overall find it improved. I still believe this paper will be of broad interest to the scientific community. That said, I still have two concerns with the paper:

Ebullition estimates: I am still concerned about the ebullitive flux estimates, and do not think these fluxes should be included in the paper. My concern for the ebullition flux estimates is as follows. The authors use floating chambers sampled every two weeks to estimate ebullition. I am not aware of any published studies that use this method as opposed to bubble traps or estimating bubbles from portable gas analyzers (the latter is tricky as the authors point out because bubbles are so spatially and temporally variable), and the authors provide no citations for this method. I don't think the floating chamber method is appropriate and I worry that publishing this method will encourage its use among researchers. The problems I see are: (1) the large surface area of a bucket (compared to bubble traps with syringes or graduated cylinders) increases the likelihood of diffusive fluxes and then reciprocally, back-diffusion, (2) the 2-week sampling period will also increase back-diffusion. In contrast, bubble traps measure volume of bubbles directly and their smaller surface area reduces diffusion in and out, and sampling often occurs more frequently. Bubble traps are also easy to use and can be deployed over long time periods (weeks to months)—this is not that logistically or financially challenging (e.g., Burke et al. 2019 JGR, Ray et al. 2023 GRL, DelSontro et al. 2016 L&O, Baron et al. 2022). The authors have made sincere attempts to justify and estimate error with their estimates, including the new supplement comparing ebullition to automated flushing chambers. However, there are still some large assumptions: (1) they assume 0 ebullition if chamber concentration is similar to water concentration (I assume there would have been some bubbling over two weeks), and (2) assume high chamber concentrations account for most of the bubbles, acknowledging back-diffusion. The flushing chamber comparison shows huge differences on the daily scale (e.g., Supplemental Table 2), and we don't know the "true" ebullitive flux without bubble traps integrating multiple days.

**We appreciate the reviewers concerns and we also have concerns about the methods and would not recommend their use if other methods were easily available. However, bubble traps suffer from a number of limitations too, as they are harder to maintain in a long-term deployment of many months. In eutrophic systems they suffer from huge biofouling and can become blocked by algal growth and filamentous algae. The reviewer is correct that very low ebullitive emission might be missed by the approach here, but the data suggest this is not a big problem as a very small proportion of the samples (28 of 401 records) were recorded as having zero ebullition. Also, we show that short-term, a few days, or a week every month of a 'better' method is a poorer estimation of ebullition in terms of % error in the estimate (median 50% error) than the continuous deployment of the static chamber (15-30% error) (supplementary materials). We cautiously revised the text to emphasize the caveats involved in their**

use, highlighting the underestimation of ebullition but we also show there is value in the data, through the comparison with the 'true' data from the flushing chamber.

**the flushing chamber does show huge daily variation with the peak in ebullition correlating with drops in atmospheric in pressure (see fig 1). These chambers do in fact measure the true ebullition as we separate the diffusive from ebullitive emission, by repeat resampling 3 minute periods and discarding periods where bubbles arrive (if R2 value is low, indicating a non-linear increase) and selecting the median beta of the regression equations that pass the test of linearity we can reliably determine diffusive flux. Then we can calculate the concentration in the chamber expected from diffusion after 4 hours and by taking this away from the final measured concentration we can separate ebullition from diffusion.**

[Figure]

.

Fig 1. Ebullitive and diffusive flux over 3 weeks period in June/July 2023, showing that peak emissions are driven by drops in atmospheric pressure (black line).

In short, these flux numbers are unreliable, and I don't think they are useful if we can't trust them (even as underestimates). As stated above, it makes me nervous to publish this method as other researchers may then use it as opposed to using bubble traps. If the authors removed the ebullition estimates, I think there is still a compelling story that periodic mixing events result in high GHG fluxes from shallow lakes.

We altered the text in the section discussing the use of chambers to the following:
"Thus, the continuous monitoring of ebullition using chambers with known biases was deemed the least

worst method available, but we acknowledge the caveat that ebullitive emissions are underestimated, (see supplementary material). We further acknowledge that this approach of static chambers should, where possible, be replaced by other methods to estimate ebullition, such as automatic flushing chambers.

Convention on assessing stratification and depth of hypolimnion: On a less fundamental, but still important, note, I still believe the authors should follow convention with defining stratification by density, which is also what should be used to identify the area of the hypolimnion (e.g., see Gray et al. 2020). The authors currently use temperature for stratification thresholds, but the physical mechanism is density, which has a non-linear relationship with temperature. Given that there is a strong temperature gradient in this study (April – October), these relationships may become important (though will not likely dramatically alter results). Further, there are no citations for using oxygen over density (or temperature) for estimating anoxia in the hypolimnion. If the authors prefer to focus on oxygen, consider using a more common metric of "anoxic fraction," or the fraction of the sediment exposed to anoxic waters (see Rabaey et al. 2023, Frontiers in Env. Science from Nurnberg et al. 1995).

**To address the concerns of the reviewer concerns we reanalyzed the data to define the thermocline depth and the area of the hypolimnion using density with the 'rlakemonitor' and 'lakeanalyser' R packages. The results are the same as our previous estimation of hypolimnetic area are not different than those we already used. We changed the methods section to include the use of these packages and methods of assessment, but the results remain unchanged.**

Introduction
• Lines 43-44: I think Rosentreter says all aquatic systems contribute to half of global emissions (not lakes and ponds specifically)?

**altered**
• Lines 57-61: As macrophytes are not important to the current study, I suggest removing.

**we are discussing controls of GHG in general in the intro so I think that it is relevant to mention macrophytes**
• Line 83: Citation was removed between drafts—add back in?

**thanks - restored**
• References to the Sondergaard paper (same lake and time frame have been removed)—it seems important to highlight this work in the Introduction, and explain how this work expands it.

Done in intro
• Lines 81-83: citation needed. Is there consensus here?

This section in well references

Methods
• Line 139: from 26 May 2020 through when in the fall?

**added**

• Line 187: was the oxidation study done in this lake? Please provide more details.

Yes – now clarified
• Lines 216-217: It is impossible to know which is "better" as we don't have true fluxes in this comparison—e.g., not compared to bubble traps.

See general comments the flushing chambers provide 'true' flux estimates as good or better than bubble traps

Discussion
• It would be helpful for the first paragraph of the Discussion to be an overarching statement of major findings and roadmap for the Discussion, rather than immediately comparing to three other studies.

We added a sentence clarifying the main aims and findings at the beginning of the discussion.

• I still find that the Discussion would be easier to follow with subheadings.

We think the discussion is clear and logically laid out, but have added subheading and would suggest the editor advises on whether they are needed.

• Lines 354-371: Other studies have also found higher ebullition linked with deeper locations in lakes, including Sø et al. 2023, Science of the Tot. Env.)

• Lines 365 – 371: See Ray et al. 2023, Sø et al. 2023—other studies examine the complexities beyond temperature

• Lines 373 – 381: Bubble traps are not that much more logistically or financially challenging – some bubble traps are deployed for long periods of time (e.g., Burke et al. 2019 JGR, Ray et al. 2023 GRL, DelSontro et al. 2016 L&O, Baron et al. 2022)—these papers would be useful to cite not just for methods, but also for interpreting ebullitive fluxes, should they continue to be included.

• Lines 450-453: for temporal sampling, the authors may want to reference Ray et al. 2023 (L&O), Natchimuthu et al. 2017 (JGR), and Wik et al. 2016 (GRL). These papers all examine temporal variability and emphasize the importance of more samples over an open-water season.

**to address the above 4 bullets we added more extensive reference including Sø 2023, and Burke 2019 and Ray 2023, DelSontro et al. 2016 to the discussion of ebullition.**

Supplement
• Table 2: Add units for flux, and note the date is DD-MM-YYYY. Caption should briefly explain timescale – what is differences from the mean? Mean floating chamber estimate of bubbles? This is also not clear in the text, and supplemental materials 2 and 3 are cited together. Please explain each individually, along with the take-home messages.
**explanations added**

**added**
Minor comments:
• I agree with Reviewer 1 of using "freshwaters" rather than "fresh waters"

Fresh waters are two words as a noun and one as an adjective, e.g. Freshwater ecology is the study of the ecology of fresh waters, which is how they are used in the paper. As this is correct English I prefer to keep it that way, but English evolves though use and it is increasingly common to see freshwaters as a noun,  so it is not a hill I am prepared to die on if you insist it is changed.

---

## Author Response (AR3)

Dear Editor and reviewers

We would like to express our gratitude to the hard work of the reviewers and editor in helping this paper towards publication. Their input has helped improve the quality of the paper and helped us better address the weaknesses in the approach. Apart from my slowness this has been an example of the peer review process working very well. Thanks again.

Yours sincerely on behalf of the authors

Thomas Davidson